Comparison of the bleaching susceptibility of coral species by using minimal samples of live corals

Wang Jih-Terng
Chu Chi-Wei
http://orcid.org/0000-0002-4371-7103 Soong Keryea keryea@g-mail.nsysu.edu.tw
Department of Oceanography, National Sun Yat-sen University , Kaohsiung , Taiwan
Waiho Khor
Electronic publication date: 2022 Jan 26
Publication date: 2022
Volume: 10
Electronic Location ID: e12840
Received 2021 Sep 1; Accepted 2022 Jan 6
Copyright: © 2022 Wang et al.
Copyright year: 2022
Copyright holder: Wang et al.
License: This is an open access article distributed under the terms of the Creative Commons Attribution License, which permits unrestricted use, distribution, reproduction and adaptation in any medium and for any purpose provided that it is properly attributed. For attribution, the original author(s), title, publication source (PeerJ) and either DOI or URL of the article must be cited.
License URL: https://creativecommons.org/licenses/by/4.0/

Keywords: Coral, Bleaching susceptibility, Photoimaging, Image analysis

Funding: Ministry of Science and Technology of Taiwan 109-2811-M-110-526 and 109-2621-B-110 -001 This project was sponsored by grants to Keryea Soong (109-2811-M-110-526) and Jih-Terng Wang (109-2621-B-110 -001) from the Ministry of Science and Technology of Taiwan. The funders had no role in study design, data collection and analysis, decision to publish, or preparation of the manuscript.

==============================
In massive bleaching events (losing symbiotic algae from corals), more sensitive corals are bleached earlier than other corals. To perform a comparison of bleaching susceptibility within and across coral species, a simple quantitative method is required. Accordingly, we present a laboratory-based method for comparing the bleaching susceptibility of various coral species by using a standardized image analysis protocol. Coral fragments were sampled from the colonies of five species selected from Kenting, southern Taiwan, and maintained in the same aquarium tank with circulating seawater; 2 seawater temperature regimes were used (i.e., fast-heating program (FHP), with a heating rate of 1 °C per day; and slow-heating program (SHP), with a heating rate of 1 °C per 3 days). Each coral fragment was photographed periodically, and the colored images were subsequently converted to grayscale images and then digitally analyzed to determine the standardized grayscale values (G0) by comparing with that of standard color strip. The G0 of a sample at each time of photographing during bleaching was divided by the difference of G0 between the acclimating and the same but completely bleached fragment to derive the relative grayscale (RG%) at a particular stage of bleaching; this is done for each coral fragment of a colony. The smaller the RG% of a coral fragment the closer it is approaching completely bleached condition. The level of decrease in RG% within a time series of images in each heating regime was used to establish a bleaching time index (BTI). The lower the BTI, the sooner to reach a defined bleaching level (e.g., 30%), this indicates the coral is more sensitive to thermal bleaching. In the experiment, we compared the bleaching susceptibility of the five species. Based on the proposed BTI, the five species were ranked in terms of bleaching susceptibility, and the rankings were identical between the two temperature regimes; three species in Pocilloporidae had lower BTI, whereas the hydrocoral Millepora species had the highest BTI. Within each heating regime, the BTI of different species were ranked and used to indicate susceptibility. In the FHP, the three Pocilloporidae species could be divided into two groups in terms of bleaching susceptibility. FHP not only displayed a higher differentiating capability on coal bleaching susceptibility than SHP, but also had a faster completion time, thus reducing the likelihood of unforeseen complications during the tank experiments. Our color-based method is easier and less effort-intensive than methods involving the assessment of zooxanthellae densities. Moreover, it requires much fewer replicates and all samples in one large tank (e.g., 300 L) for the studies considering multiple species comparisons. This method opens opportunities for studying the effects of species types, acclimatization (e.g., seasons), and environmental factors other than temperature on coral bleaching.

Introduction

Coral bleaching is primarily caused by the breakdown of coral-algal symbiosis under heat stress; the expulsion of symbiotic algae from thin coral tissues reveals the white carbonate skeleton underneath (Fitt et al., 2001; Douglas, 2003; Lesser & Farrell, 2004). In the context of global warming, the substantial spatial scale of coral bleaching with high mortality rates has caused a worldwide decline in coral coverage that has had a destructive impact on reef ecology (Hoegh-Guldberg et al., 2007; Jones et al., 2008). The increasing incidence of beyond-threshold seawater temperatures has increased the frequency of coral bleaching events such as that at the Great Barrier Reef in 2016 (Hughes et al., 2018). The large scale of coral bleaching highlights the catastrophic impact of global warming; this has thus stimulated considerable attention on the preservation of coral reefs.

Although the mechanisms underpinning coral bleaching are still not fully understood (reviewed in van Oppen & Lough, 2018), numerous efforts have been devoted to explore the possibility of enhancing the capacity of corals to survive at higher (future) temperatures. Strategies proposed for improving coral resistance to thermal stress entail applying temperature acclimatization and adaptation measures (e.g., Bellantuono, Hoegh-Guldberg & Rodriguez-Lanetty, 2012; Putnam & Gates, 2015; Majerova et al., 2021), searching for heat-tolerant genotypes in natural or artificial high-temperature environments (e.g., Coles & Riegl, 2012; Kao et al., 2018), applying probiotic microbe inoculation (Rosado et al., 2019), and using CRISPR/Cas9-editing techniques to mediate heat tolerance capability (Cleves et al., 2018). These strategies require examining the bleaching performance of corals in tank experiments under a temperature-increasing process to verify the thermal tolerability of the treated corals.

The rate of decrease in algal density in coral tissues is conventionally used as a measure to directly demonstrate the bleaching level of corals under thermal stress. Algal cell density, occasionally combined with chlorophyll a content from corals (e.g., Hoegh-Guldberg & Smith, 1989) or used along with a detailed measurement of photosynthetic processes and related physiological parameters (e.g., Jones et al., 1998; Fitt et al., 2001) enables monitoring the physiological condition of individual coral colonies under stress. However, deriving this measure necessitates specialized equipment and considerable effort, thus restricting the application of this approach. In addition to cell-counting methods, observer-based methods have been developed for field surveys of coral health through the use of color reference cards with multiple hues (Siebeck et al., 2006) or use of simple color reference strips supplemented with photographic images (Winters et al., 2009; Chow et al., 2016; Bryant et al., 2017). The feasibility and reliability of these methods have been confirmed using various approaches (e.g., Montano et al., 2010; Amid et al., 2018). Observer-based methods that rely on a subjective color-based assessment of corals are quicker and less expensive than cell-counting methods but are potentially biased by the personal judgements of assessors. Such bias, however, could be addressed by applying images captured under a set of fixed conditions and then analyzing the images through computer software. Photography-based methods have considerably improved the operational efficiency of field surveys of coral bleaching events. Moreover, studies have demonstrated that image data were highly correlated with measurements of the algal cell density and chlorophyll content of corals, confirming the appropriateness of using image data to research bleaching conditions (Winters et al., 2009; Chow et al., 2016; Bryant et al., 2017). However, studies have yet to apply photography-based methods in tank experiments. Instead, they have generally applied cell-counting methods to compare the bleaching susceptibility of corals with different acclimatization histories or experimentally treated coral samples; such methods require a considerable quantity of living corals and more sophisticated handling efforts and procedures.

To fill this gap, the present study applied a photography-based method to distinguish coral bleaching under different conditions, such as different species and heating regimes, in a tank. We developed a bleaching time index (BTI), calculated from the changes in relative monotone grayscale of images with time, to compare coral bleaching susceptibility.

Materials and Methods

A total of five species of corals from the shallow waters of Kenting, southern Taiwan (120°41′43″E, 21°59′5″N), were collected by scuba divers. The collection permit for this investigation was issued by the Kenting National Park Authority of Taiwan to KS. Four scleractinian corals (Seriatopora caliendrum, Pocillopora verrucosa, Pocillopora damicornis, and Favites complanata) and one zooxanthellate hydrocoral (Millepora intricata) were included in this study. For each coral species, five colony replicates were selected, and two coral fragments (4–8 cm per fragment) from each colony were harvested for the bleaching experiment. The species were transported to the laboratory and separated into two identical aquarium tanks, with each containing 300 L of seawater. Thus, each species contained five colony replications for each of two treatments, and the coral fragments of all five species were maintained in the same tank, for each temperature treatment.

Figure 1 illustrates the procedures of the study protocol. Specifically, the coral fragments harvested from all colonies of the five coral species were acclimatized at 27 °C in one of the tanks for 1 week prior to the experiment. The seawater in the aquarium tank was first circulated horizontally by two underwater pumps—each of which provided a flow rate of 5,200 L h−1—and then filtered through several layers of aquarium filter pads using another pump (flow rate: 2,000 L h−1). The corals were illuminated with photosynthetically active radiation (25–30 μmol m−2 s−1) under a 12:12-h light–dark regime. Heating of seawater was conducted with two heaters (500 w for each) controlled by a digital thermostat (ISTA tsb-958), which were located right in front of an underwater pump to quickly mix water and away from tested corals. During the heat treatment, one of the tanks was used to implement a fast-heating program (FHP; temperature increment rate = 1 °C per day), and the other tank was used to implement a slow-heating program (SHP; temperature increment rate = 1 °C per 3 days). The temperature setting was changed at noon each day, and approximately 1 h of heating was generally required to reach the target temperature. The water temperature was raised to 35 °C and maintained at that level until the entire coral fragments were pale.

Figure 1 Outline of the study procedure for determining bleaching time index (BTI) of corals through image analysis.

During the heat treatment, one image of all colony replicates of each of the 5 species was captured twice a day (at 6:00 AM and 6:00 PM) under a constant photographing condition. All images were captured by an Olympus Tough TG-5 camera (Olympus, Tokyo, Japan) with an attached LED ring light (Weefine, Guangdong, China), aperture speed of 1/50 s, ISO level of 100, and aperture size of f/2.8; the camera was operated in macro mode, with the built-in camera flash turned off. The image acquisition process was completed when the coral fragments turned completely pale (i.e., no grayscale decreasing in 6 consecutive data points). All images were stored in JPG format for further bleaching level calculation.

The bleaching level of the coral fragments during the heat treatment was determined by deriving the changes in the relative grayscale values of a fixed square area in the images captured at every time point, as illustrated in Fig. S1. Adobe Photoshop CC (2017) was used to measure the grayscale value in each target square area of coral sample (GC) and standard color strip (GB for black and GW for white). Before measuring the grayscale of a selected area, the color image was firstly converted to grayscale, and then, that of the selected area was averaged with the Photoshop software. To correct for the variations in conditions between image acquisition sessions, the GC value derived for each image captured at each time point (6:00 AM and 6:00 PM) was first standardized with GB and GW to derive the standardized grayscale (G0), as presented in Eq. (1). A step-by-step procedure for estimating G0 was provided in Fig. S2. Subsequently, to measure color loss at each time point, the remaining color intensity, derived as G0−G0bleach, was normalized with the total grayscale change ( G0normal−G0bleach) to derive the relative grayscale (RG%) of a sample at each time point during bleaching, as presented in Eq. (2). G0bleach represents the average G0 of bleached coral fragments calculated from the images captured at the last 5 time points of heating experiment when the grayscale of coral fragment displays not further decrease; and G0normal was derived from the mean of the first 5 G0 data points at 27 °C incubation. All preceding calculations were conducted for each fragment of each species; that is, the calculations of relative changes were affected only by the performance of the coral fragment.

(1) G0=(GC−GWGB−GW)

(2) RG%=G0−G0bleachG0normal−G0bleach×100%

The calculated RG% values were plotted against incubation time and analyzed through curve fitting in SigmaPlot 14.1 software. The best-fitting regression analysis indicated a modified Gaussian regression model revealing the highest regression coefficient (R2). Parameters and R2 values derived using the modified Gaussian equation for all 50 data sets are presented in Table S1; the R2 values ranged from 0.9326 to 0.9978 (mean ± SD = 0.9856 ± 0.0125). The modified Gaussian regression equation was further used to estimate the time (in days) required for the grayscale to decrease during bleaching, and then applied to develop the BTI for evaluation of the bleaching rate and heat stress susceptibility of the various coral species. With the regression equation, SigmaPlot provided the corresponding X (incubation time point) and Y (RG%) values within the range of input data, which were used to calculate the incubation time (in days) at which RG% decreased by 10%, 20%, 30%, 40%, or 50% through interpolation. Because 10%, 20%, 30%, 40%, and 50% decreases in grayscale (color) values produced similar results in terms of the differentiation of bleaching susceptibility among the species, only results obtained at 10%, 30%, and 50% are presented in this study. We defined the time required to reduce the RG% value of coral samples during the heat treatment as the BTI. BTI values were compared between the coral species by using one-way analysis of variance followed by Tukey’s test for post-hoc analyses. The appropriateness of using BTI to evaluate bleaching susceptibility was also examined with an analysis by Pearson correlation coefficient between the treatments of FHP and SHP.

Results

When seawater was gradually heated to and maintained at 35 °C in both the FHP (1 °C per day) and SHP (1 °C per 3 days), the bleaching responses of the coral fragments harvested from the colonies of the 5 coral species varied (Fig. 2B–2F, 3B–3F, and 4. In Figs. 2 and 3, we considered a 5% decrease in relative grayscale as the criterion to calculate the BTI in order to determine the time at which initial bleaching occurred. Accordingly, we observed that in the FHP, coral bleaching started on day 3.2 ± 0.3 for S. caliendrum, day 5.5 ± 0.4 for P. verrucosa, day 6.4 ± 1.1 for P. damicornis, day 6.8 ± 0.9 for F. complanata, and day 8.4 ± 0.8 for M. intricata. In the SHP, the bleaching started on day 7.9 ± 1.6 for S. caliendrum, day 9.9 ± 4.2 for P. verrucosa, day 7.9 ± 3.2 for P. damicornis, day 12.2 ± 2.6 for F. complanata, and day 22.5 ± 2.2 for M. intricata. Figures 2B–2F and 3B–3F also showed higher within-species variations in the days of starting bleaching as treated by SHP (CV%: 10.0–40.5%) than by FHP (CV%: 6.8–17.2%) in the five testing coral species. Notably, the susceptibility rankings of the coral replicates were the same between the FHP and SHP.

Figure 2 Bleaching responses of corals in the fast-heating program (FHP, 1 °C per day).

(A) Temperature increment program, in which the seawater temperature was increased at a rate of 1 °C per day from 27 °C to 35 °C and maintained at 35 °C until the end of the experiment. (B) Seriatopora caliendrum, (C) Pocillopora verrucosa, (D) Pocillopora damicornis, (E) Favites complanata, and (F) Millepora intricata treated in one 300-L aquarium tank. Five color symbols represent coral fragments from different colonies.

Figure 3 Bleaching responses of corals in the slow-heating program (SHP, 1 °C per 3 days).

(A) Temperature increment program, in which the seawater temperature was increased, after 3 days of acclimation, at a rate of 1 °C from 27 °C to 35 °C and maintained at 35 °C until the end of the experiment. (B) Seriatopora caliendrum, (C) Pocillopora verrucosa, (D) Pocillopora damicornis, (E) Favites complanata, and (F) Millepora intricata treated in one 300-L aquarium tank. Five color symbols represent the coral fragments from different colonies.

Figure 4 Bleaching time index (BTI) values calculated for the five coral species in the fast-heating program (FHP) and slow-heating program (SHP).

The decrease in the relative grayscale for calculating BTI at the cutoff values of 10% (A and B), 30% (C and D) and 50% (E and F) was derived from the data (n = 5) in Fig. 2 (FHP) and Fig. 3 (SHP). Whisker caps: the highest and lowest values; box: 95% confidence intervals; black line: medium; red line: mean. Boxes labeled with the same letter are not significantly different at p = 0.05 (Tukey’s post hoc analysis, n = 5). Sc: Seriatopora caliendrum; Pv: Pocillopora verrucosa; Pd: Pocillopora damicornis; Fc: Favites complanata, Mi: Millepora intricata.

In Figs. 2 and 3, the hydrocoral M. intricata also exhibited the highest tolerance to temperature increases compared with the other four scleractinian coral species; specifically, it exhibited the longest endurance before the initiation of bleaching, tolerated the highest temperature before the initiation of bleaching, and exhibited the slowest bleaching rate. In addition, M. intricata displayed on average >40% increase in the relative grayscale before the coral started bleaching in the SHP, compared with ≦10% in the FHP and in both FHP and SHP for the other scleractinian coral species (Table S2).

Further to compare the BTI at different levels, as shown in Fig. 4, the five tested coral species displayed significantly varied response to the two treatments (p < 0.001, see ANOVA results in Table S3). However, the rankings of bleaching resistance among species determined by BTI10%, BTI30%, and BTI50% were identical in both heat treatment programs. A comparison of the 95% confidence intervals for the FHP and SHP (Fig. 4) revealed that eight out of 10 possible species pairs could be significantly distinguished in terms of bleaching resistance in the FHP; however, only 6 of 10 pairs could be significantly separated in terms of bleaching resistance in the SHP. A higher BTI indicates a greater bleaching resistance. Hence, in the FHP, the 5 coral species could be ranked (in descending order) into 4 groups as follows according to their bleaching resistance: M. intricata > F. complanata ≧ (P. damicornis = P. verrucosa) > S. caliendrum (Fig. 4 (ACE)). However, in the SHP, the species could be ranked (in descending order) into only 3 groups as follows according to their bleaching resistance: M. intricata > F. complanata > (P. damicornis = P. verrucosa = S. caliendrum) (Fig. 4 (BDF)). Furthermore, the bleaching performance of the five coral species treated in the FHP were also significantly correlated with those of the species treated in the SHP (Fig. 5). The Pearson correlation coefficients (r) for the BTI data between the SHP and FHP were 0.754 for BTI10% (p < 0.01), 0.895 for BTI30% (p < 0.01), and 0.898 for BTI50% (p < 0.01).

Figure 5 An example plot of the correlation between BTI obtained in the slow-heating program (SHP) and fast-heating program (FHP).

This example plot was derived from the results of BTI30%. Pearson correlation coefficient = 0.90 (p < 0.01).

The variation of BTI method was further examined with the coefficients of variation (CV%) of BTI values derived from colony replicates of each species. As indicated in Table S4, the CV% levels of the BTI values at the 10%, 20%, 30%, 40%, and 50% cutoff levels were low in all the tested coral species treated by the FHP (mean ± SD = 8.3% ± 3.7%; n = 25) and two less sensitive species, namely F. complanata and M. intricata, by SHP (9.3% ± 4.0%; n = 10), but high in three thermal-sensitive species, namely S. caliendrum, P. verrucosa, and P. damicornis, by the SHP (24.7% ± 10.1%; n = 15). In summary, the results indicated that SHP would result in higher within-species variation in the BTI method when applied to heat sensitive coral species.

Discussion

This study proposes a color-based protocol that entails the use of the BTI to evaluate coral bleaching susceptibility in tanks. The BTI is calculated from the changes in the relative grayscale values of a selected area in images of corals. In this study, the BTI10%, BTI30%, and BTI50% values (Figs. 4A, 4C and 4E) obtained in the FHP could be used to rank the 5 coral species into 4 groups of bleaching susceptibility. In contrast to the rankings observed in the FHP, the species treated in the SHP could be ranked into only three groups according to their bleaching susceptibility (Figs. 4B, 4D and 4F). Though the rankings were consistent between the SHP and FHP, slow heating rate did not increase resolution in differentiating bleaching susceptibility between the coral species. Consistent ranking between the SHP and FHP demonstrated that the proposed BTI can be a fair tool used to assess natural processes. Besides that, monitoring bleaching process by the SHP displayed two features different from that by FHP. First, SHP resulted in higher within-species variations in estimating BTI values of 3 heat sensitive species, S. caliendrum, P. verrucosa, and P. damicornis than FHP did (Table S4). Within species, different susceptibility levels may be caused by both genetic and/or local environmental factors, which might be only observed by SHP. However, to the heat sensitive species, the likelihood of unforeseen complications occurring in tank experiment also can’t be ruled out due to longer incubation time in tank condition. Second, to heat resistant species, like hydrocoral M. intricata in this study, there was a higher percentage of increase in relative grayscale in the SHP (40% in average) than in the FHP (10% in average) before it declined relative grayscale below 100% (Figs. 2F and 3F, Table S2). The increases in relative grayscale in the initial period of bleaching trials might be due to the accumulation of expelled symbiotic algae in the topical area of coral tissue. The high initial rise of RG% of M. intricata might be partly attributable to their heat resistant nature resulting in a slower mode of symbiotic algae releasing, and partly to their porous skeleton which made the releasing of symbiotic algae more slowly.

Based on our results, the FHP are superior to the SHP in comparing the bleaching susceptibility coral species. Nevertheless, it remains to be tested whether the BTI could also produce the same rankings for species subjected to other methods of bleaching induction. The possible effects of zooxanthellae, feeding, and light can also be tested under laboratory conditions using this method.

The bleaching susceptibility levels of the scleractinian coral species observed in this study using the proposed BTI are comparable to those reported by previous studies (e.g., Loya et al., 2001; Keshavmurthy et al., 2014); however, the results observed for the branching hydrocoral coral M. intricata are not comparable to those in the literature. Previous studies conducting field surveys have indicated that Millepora species are thermally sensitive (Loya et al., 2001; Dias & Gondim, 2016; Teixeira et al., 2019; Duarte et al., 2020), but we discovered that M. intricata was the most thermally resistant species in this study. This conflict might be derived from two possibilities, method bias and/or the other biological variations. The essential difference between BTI method and field survey on M. intricata is that the former provided time-series quantitative observations of the same coral fragments and that the bleaching was relative to the acclimation periods. On the contrary, the evidence of heat sensitive in Millepora coral was all based on visual observation during field surveys. Field survey data have been limited to snapshots of multiple species taken at the same time; the assessments of bleaching were qualitative, although many colonies were selected for sampling. Of course, we cannot exclude other biological factors that might affect coral bleaching susceptibility, such as thermal sensitivity of algal symbionts (Sampayo et al., 2008; Stat & Gates, 2011; Howells et al., 2012; Hsu et al., 2012; Silverstein, Correa & Baker, 2012; Keshavmurthy et al., 2014), tissue thickness and stress enzyme activities of the coral host (Loya et al., 2001; Fitt et al., 2009; Wang et al., 2019), with or without probiotic microbes association (Rosado et al., 2019 and the references therein), and life histories with or without acclimatization or adaptation to higher temperature (Howells et al., 2016; DeCarlo et al., 2019; Wang et al., 2019; Barott et al., 2021). More studies are required to clarify the effects of these factors, and we suggest the methods developed here is more effective and efficient.

Imaging methods have several advantages over commonly used cell-counting methods (Table 1). First, imaging methods do not require sacrificing coral samples for cell counting; therefore, increasing sampling intervals does not require proportionally increasing the number of coral fragment samples, as is the case in cell-counting methods. For example, if our experiment had been conducted using a cell-counting method, 60 times of replicate of coral fragments (i.e., 300) would have been required for each species to generate the same number of observations.

Table 1 Comparisons of imaging and algal cell-counting methods for quantifying coral bleaching among species.

Requirements and advantages	Methodology	
Counting algal cells (CZ)	Photography image (PI)	
Sampling area	Hard to obtain accurate area, especially for branching corals	No need to calculate sampling area	
Number of nubbins per colony	Number of nubbins per colony needs to increases as number of observations increase from the beginning of the experiment to the end of bleaching	Same coral nubbins for multiple times of non-destructive photography. Sample numbers could be easily increased without collecting more/larger coral fragments	
Equipment	Water pick, centrifuge, microscope, hemocytometer etc.	Camera, light, color card, software, e.g., Photoshop etc.	
Skills	Cell-biology training	Digital photography training	
Lab facility	Proportionally larger space according to: species numbers, and numbers of observation per spp.	Proportionally larger space according to species numbers, only	
Raw data	Y: Cell densities	Y: % in grayscale	
Sampling assumptions	Homogeneity among nubbins within a colony in cell numbers	No such assumption is needed since the same nubbins were observed continually	
Numbers of data point	Constrained by the numbers of nubbins available for assay	Flexible; depending on intervals between shots	
Numbers of species compared simultaneously	Less	More, we did 5 species comparison in a 300 L aquarium tank	

Second, due to much smaller sample sizes required for a species, many species could be accommodated in one tank. This effectively avoided the tank effect, which we are not really interested any way, and also reduced the potential complication of pseudo-replication in experimental design (Cornwall & Hurd, 2016).

Third, imaging methods have lower technological requirements and are less labor intensive than cell-counting methods. Unlike an underwater imaging method used a decade ago (Winters et al., 2009), current imaging approaches are easier with the increased availability of automatically controlled underwater cameras, such as that used in this study. Furthermore, compared with labor-intensive cell-counting methods, imaging methods provide additional data points at low marginal costs.

Fourth, imaging methods provide a lower level of data bias than do cell-counting methods when used for intertaxa comparisons of bleaching performance under the same controlled conditions. Cell-counting methods usually underestimate algal densities in coral species with porous skeletons, such as Millepora spp.; in such species, complete recovery of algal cells from coral specimens by using methods commonly applied to scleractinian corals (e.g., water-pik or air brush; Edmunds, 1999) is almost impractical. By contrast, the imaging method used in this study derived only monotone (or grayscale) color changes during the bleaching process, which can provide the same observation of this process as the naked eyes. Although the colors of noncoral backgrounds (e.g., that of endolithic algae or the supporting materials for coral fixation) may distort the actual grayscale levels of coral samples in an image, such backgrounds can be removed from the images during the calculation of relative grayscale values within a sample. Therefore, our protocol affords more freedom in the selection of a region of interest in an image for calculating grayscale values, which reduces the number of coral fragment replication points to one. This is an advantage over the protocol proposed by Chow et al. (2016), in which more than 100 coral fragment areas are required for calculating grayscale values.

Imaging methods, nevertheless, have disadvantages in the monitoring of coral bleaching because they require a series of images, including images of the normal and the bleached stages of corals. Consequently, imaging methods are not suitable for directly assessing bleaching conditions in nature. Also, converting JPG image into grayscale is a nonlinear compression process, which might affect greyscale quantification. However, according to our results, this defect might only have minor impact on the final comparisons.

Conclusions

In summary, we developed a BTI for comparing the bleaching susceptibility of corals based on imaging method. The method used coral fragments efficiently and the small numbers of fragments themselves allow multiple species comparisons in the same tanks. This characteristic enable investigations of a wide variety of factors causing coral bleaching in the laboratory.

Supplemental Information

Supplemental Information 1 Arrangement of coral samples for image collection.

Numbers in white boxes represent colony replicates. The region of interest in the image for grayscale calculation is indicated by the black square, and the same region was used to calculate grayscale changes in coral fragments during heating experiment with FHP (ACEGI) and SHP (BDFHJ). (AB): Seriatopora caliendrum; (CD): Pocillopora verrucosa; Pocillopora damicornis (EF); (GH): Favites complanata; (IJ): Millepora intricata. Color strip in the middle of each photograph was used to standardize the grayscale value of each target area.

Click here for additional data file.

Supplemental Information 2 Step-by-step procedure for estimating grayscale in a selected area of coral fragment.

Click here for additional data file.

Supplemental Information 3 Regression coefficients and parameters of the regression equations calculated from 5 colony replicates of each coral species.

The relative grayscale values of each coral replicate during heat treatment were fitted onto a modified Gaussian equation to describe the bleaching process. Modified Gaussian equation: f(X) = y0 + a × exp( − 0.5 × abs[(X − x0)/b]c).

Click here for additional data file.

Supplemental Information 4 The levels of increase in relative grayscale of a coral fragment during heat treatment.

The data were calculated from subtracting 100% by the highest relative grayscale obtained during heating process, and expressed as Mean ± SD (n=5).

Click here for additional data file.

Supplemental Information 5 One-Way ANOVA of the response of coral species to heat treatments.

Click here for additional data file.

Supplemental Information 6 The within-species variations for BTI measurement in coral bleaching by fast heating program (FHP) and slow heating program (SHP).

The variations were estimated by coefficient of variations (CV).

Click here for additional data file.

Supplemental Information 7 Raw data for Fig. 2 & 3 (relative gray scale for 5 species by days).

Click here for additional data file.

Supplemental Information 8 Raw data for Fig. 4 and 5 (BTI values for 5 species).

Click here for additional data file.

We thank Te-Yu Chen, Chung-Wei Yang and Shannon Huang for assisting with the field and lab work, and Prof. Allen CA Chen for the identification of coral species. We also like to thank Paul Hussain and Yankuba Sanyang from Wallace Academic Editing for their assistance in English editing. Dr. Edson Vieira and another two anonymous reviewers provided many valuable comments and suggestion, which are greatly appreciated.

Additional Information and Declarations

Competing Interests

Author Contributions

Field Study Permissions

Data Availability

The authors declare that they have no competing interests.

Jih-Terng Wang conceived and designed the experiments, performed the experiments, analyzed the data, prepared figures and/or tables, authored or reviewed drafts of the paper, and approved the final draft.

Chi-Wei Chu performed the experiments, analyzed the data, prepared figures and/or tables, and approved the final draft.

Keryea Soong conceived and designed the experiments, analyzed the data, authored or reviewed drafts of the paper, and approved the final draft.

The following information was supplied relating to field study approvals (i.e., approving body and any reference numbers):

Collection of coral was approved by Kending National Park.

The following information was supplied regarding data availability:

The raw measurements are available in the Supplemental Files.

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
