# Peer review of "Comparison of the bleaching susceptibility of coral species by using minimal samples of live corals"

_PeerJ, doi:10.7717/peerj.12840_

## Round 0.1 · original submission · Major Revisions

Dear authors, I do agree with the reviewers that overall, this is a nicely written manuscript. Therefore, I would like to invite the authors to kindly address the concerns of the reviewers during your revision.

·

Basic reporting

The paper is pretty well written, the references are adequate and the hypothesis is clear.

The structure is overall ok, however several parts of the paper (see additional comments) need rephrasing, more information and justification (specially in the methods) and more clarity in the results interpretation and discussion.

Data were made available as supplementary material.

Experimental design

The research is original, important and very interesting, specially by proposing a feasible and applicable method to evaluate coral bleaching, which is a serious issue all over the worlds.

Sadly, there are a lot of information missing in the methods (such as experimental conditions and the relationship to natural field conditions), which are very important to better understand, replicate and ground the study.

Besides, there are a very confusing factor in the study design (variable colored background among species and between heat treatments) that should be performed different or at least deeply explained to convince the reader that it is not a great flaw of the study.

Validity of the findings

The results are very interesting, clear and important.

However the way some of them were obtained is not clear, and also some interpretation is not possible to be made by the reader because it is not clear how authors get to them as they seem not to relate to one or more of the figures. Besides, some results do not have a clear point of what they mean or why they are important.

The discussion needs a great restructuring because some interpretation are not clear in terms of how they were performed or why they are important. Additionally, several times authors discuss some points, although very important, that do not seem to relate to their study or results.

The conclusion is pretty clear though and I suggest to the authors to focus on it to build their discussion and select the necessary results that need to be shown and interpreted to this study.

Additional comments

Dear authors, I really appreciate to read the manuscript and the main ideas investigated in the study. The proposed method for easily evaluating bleaching through time under different conditions or among different species is really interesting, important, and feasible. However, there are some issues that need to be addressed in order to provide a better and understandable manuscript: 1) a lot of information that is needed to understand how the experiment conditions mimic those in the field is missing; 2) the colored background in the photos, which varied among species and between heat programs, may interfere in the results, which I think it is a great flaw if not properly explained/justified; 3) several results are not clear on how they were interpreted or what is their relevance; and 4) the discussion also need more work in the structure and on better connections and interpretation of the necessary results instead of general discussion about some other important related topics but that were not originate in the results of the study.

Detailed Comments:

Abstract
L24-26 – After reading the paper I realized that authors did a standardization of the greyscale by setting the grey-scale value of a given nubbin during acclimation as 100% and the grey-scale value after bleaching as 0%. It is important to rephrase this part to let it clear that 100 and 0% corresponds to a standardization and not to the real grey-scale values, which would mean total black for 100% and total white for 0%.
L31-32 – This part brings some confusion because short/long bleaching time may give the idea of time necessary to bleach (which I think is the correct) or the duration of bleaching. I suggest rephrasing it properly.
L33-34 – Why? What does that mean? Was it a decision that emerged from the experiment or something that the authors had imposed?
L34-36 – I agree that the proposed method for bleaching evaluation is better than quantifying zooxanthella density because it is easier and less time-consuming. And that it requires less replicates for studies considering several data points collected in different times. However, this must be clear here. Otherwise, the reader may think that the authors are talking about accuracy and precision of the different methods.

Methods
L104-106 – What were the total replicate? I guess it was 5 species x 5 colonies x 2 nubbins, totalizing 50 replicates, is that right? Were half of them assigned to one of the experimental tanks, so 5 nubbins of each species per tank? This information must be made clearer.
L108-110 – Authors should use ‘mol’ instead of ‘E’, once the first is part of the SI and the second is not. How does this PAR value compare to natural PAR in the field? Additionally, was any dawn/dusk simulation performed? These are important information to be mentioned because bleaching response could be triggered by the wrong amount of PAR reaching the corals. Since the proposed method is to be used in studies with an aim of selecting resistant corals or comparing different species susceptibility, experimental conditions must be as similar as possible to natural ones.
L110-114 – How was temperature increase performed? Thermostats? Where were they inside the tank? This is important to let it clear if some nubbins/species were closer to the heating source when compared to others.
L114-115 – 35 °C is a considerable high temperature. What are average, maximum, and minimum temperatures in the field? As for the PAR (mentioned before) temperature in the field is also important to be mimic in the lab to obtain more reliable results. If 35 °C is not a possible temperature to be reached in the field or considering future projections, authors must justify their choice of heating up to 35 °C.
L121-122 – I would add here a couple more words letting clear to the reader that it is because of this criterion (stop taking pictures when the entire nubbin was pale) that for each species time range with data points in Figs 2 and 3 differ among species.
L125-127 – It was surprising for me that the grayscale was estimated considering the square surrounding area instead of the exact coral area. That brought some questions to me. First, how was the greyscale estimated in Photoshop? Was it an average value considering all the pixel inside the squared area? A value of what? Does the software return somehow this average value? This information would be interesting to be mentioned in the main text for further studies using this approach. Second, the squared area instead of the exact area would be a lesser problem if all the plastic backgrounds were the same color, but even though, it would add differently to the final grayscale value depending on the coral nubbin shape. However, as shown in the supplementary figure, the background color varied both among species in a given heat treatment and also between heat treatments for a given species. Clearly each background color returns a different grayscale value, and this probably add some noise to the coral color estimate. The question is, how much noise does it add and how much that noise may change the final results?
L147-148 – Why not using a two-way ANOVA to insert the heat program as a factor as well? One of the goals of this study is to compare the heat programs, so they would be formally included as a factor in the analysis.

Results
L156-164 – It all seems methods to me.
L166-171 – Call Fig 2.
L171-173 – How did authors interpret these results? In which figure or table could the reader reach the same interpretation? Thus, I suggest to clearly refer to the figure necessary to the reader to get to the same conclusion.
L180-185 – Again, this part fits better in methods than in results.
L185 – …bLeaching…
L185-187 – How did the authors concluded that? Was there a formal test? Examining Fig 4, it seems that there are great differences among heat treatments for a same species considering the different BTI. This is particularly true considering that the y-axis scale is different for each heat treatment (maximum of 18 for FHP and maximum of 30 for SHP).
L187-190 – Was this visually or formally evaluated with some statistical analysis? If yes, where are the results? Additionally, the one-way ANOVA tables comparing the BTI among species in a given heat program was not shown.
L196-199 – What does this result indicate biologically? Is it relevant somehow? If yes, authors should explain this rationale in the methods, besides saying that the regression was performed, which it is not mentioned before. Additionally, why is only BTI30 shown in Fig 5?
L200-206 – This part is very confusing and I did not get what this result indicates.

Discussion
L212-223 – Although I agree with the idea, it has nothing to do with the study. The main results compare different species and do not evaluate colony-based variation.
L223-227- This is a very confusing part, and I did not get the point of it. Why is the SHP ‘superior’ than the FHP when comparing bleaching susceptibility if they return similar results? Actually, the SHP return only 3 groups on the susceptibility ranking, therefore, FHP seems more adequate to me. Actually, the authors end this paragraph saying that FHP is better, however without grounding this conclusion considering the previous statements. I suggest a complete rephrase and restructure of this paragraph. Besides, I suggest to move the discussion comparing heat programs to after the paragraphs discussing the comparison of susceptibility among species, it seems more logical and aligned with the aims of the study.
L228 - … of the STONY CORAL species…
L228-241 – I really appreciated the comparison with the literature, this is important. However, I missed more discussion about why the fire coral may be more resistant and why it actually even increases the greyscale value at some point during the experiment.
L242-248 – I did not understand why authors here discuss other factors that may modulate bleaching susceptibility if they have not yet discussed any factor that could explain the susceptibility rank obtained in this study (see previous comment about the previous paragraph).
L255 – merge with the previous paragraph.
L268-271 – Was it done in this study? How? If so, it wasn’t mentioned in the methods.

Figures
- It would be very useful to have the name and a small photo of each species in the respective graph in Fig 2, 3 and 4.

Reviewer 2 ·

Basic reporting

Please see the following comments:
Line 19: Nubbins- I strongly suggest using the word fragment instead throughout the text.
Line 25-26 - Unclear sentence: The average grayscale obtained for the samples during the acclimation period was 100%, whereas those obtained for the samples after bleaching were 0% for each species.
Line 31- Millepora is not a scleractinian coral, maybe distinguish in the abstract.
Line 35- zooxanthella- Add e throughout the text
Line 40-49- add to abstract what is bleaching?
Line 94- I suggest distinguishing more between scleractinian corals and “hydrocoral”.
Line 104- illustrates THE procedures
Line 122: did you use the RAW images? This is important to say here.
Line 137- This thus. Remove thus, explain more on the problem?
Line 149-151: can you please elaborate on this point?
Line 206- I suggest to add one sentence summary of the results
Line 212-213: Perhaps add another reference to strengthen this statement?
Line 219: does this not contradict the previous paragraph?
Line 223: FSP? Not FHP?
Line 225-227: not clear. Why is the FHP superior?
Line 241: what do you mean, assessments by divers in Situ?
Line 274: Consider adding a figure of the workflow on the computer- how you sample an area of the fragment?
Line 285: Adaptation or acclimation?
Line 287: Please correct the sentence

Figure 2 and 3: consider adding titles to the subfigures. Also, in the text state clearly if there was more variance between colonies or between treatments. This is mentioned, but needs to be more clear.
Figure 4 caption: reverse sentence, try decrease in the relative grayscale, based on… Also consider adding the number of fragments n for box plots.
Figure 5: caption is missing the conclusion of the figure.
Supplementary figure S1 needs a caption or to add on the image the coral names
Table 1: last line jumped (page 27 on PDF)

Experimental design

I strongly suggest to use RAW images in this protocol, which is not stated in the text.

Validity of the findings

See comments on section 1 (Basic Reporting).
I strongly suggest to add a figure of the workflow on the computer- how do you use a gray scale image and subsample pixels. This will be very helpful for those who wish to employ this method in their research.

Additional comments

Dear authors. I enjoyed reading the manuscript. However there are some grammatical issues as well as semantics such as the use of the word nubbins. Personally, I find automatic grammar correction tools such as Grammarly very useful in checking final drafts before submission.
I look forward to reading an improved version of this manuscript and seeing this paper published.

Reviewer 3 ·

Basic reporting

This is a well written paper with an appropriately detailed methods and results section. It provides context on both the ecological importance of coral blecahing/color change and the development of techniques to measure it, alongside their strengths and weaknesses. The manuscript has appropriate figures.

Experimental design

The authors use the first several time points as the control for their color scores and, I believe, have a single tank replicate for each temperature control (this could use some clarification). While i agree that this is probably an acceptable approach for the type of methods development they are doing, I think it would be appropriate to more explicitly address their lack of tank replication and a baseline control.

It would also be nice to see (in text or figure) that there is a stable (absolute? relative?) grayscale during the acclimation period, both to ensure that a week was appropriate and that the use of these in place of a proper control is approriate.

Validity of the findings

I have one primary question about a detail that may actually be importnat for the overall direction of the manuscript. Why is there an increase in figure 3f (m. intricata)? The authors mention this in the results, but what is the actual cause of this likely to be and what does it mean for the use of this method? I also think the wording on line 179 ('period of acclimation') is confusing since it appears during this point the corals were reaching max temperature, not acclimating.


The conclusions and discussion of this paper are appropriate and are nicely matched with the figures. The only thing i would like to see the authors add is a bit more discussion of the fact that the SHP can likely be used to select genotype level differences (different colored points), making it a better approach and strengthening the utility of this method. If so, the authors may elect to add some analysis of this, which i think would benefit the paper and its citability.

---

## Round 0.2 · Minor Revisions

As noted by the reviewer, I agree that the current revised manuscript improved significantly. I hope the authors could kindly address the small concern raised by the reviewer in the matter of image acquisition accordingly.

Reviewer 2 ·

Basic reporting

The manuscript was improved on these points.

Experimental design

From my prior comments:
did you use the RAW images? This is important to say here.
Re: We used jpg files as correction against color cards and our formula render everything relative. Using RAW does not have the benefits of giving true color anyway, this is especially true when the method is employed by different people/teams. We chose calibration against the same color card, this is more practical and is also easy to use.


I still think it is important to note that you used JPG. It is a nonlinear compression that affects the grayscale values. It is still acceptable but I strongly suggest to mention this in the discussion.

Validity of the findings

Figure S2 helps to replicate this study.

Additional comments

Dear authors, you have substantially improved the manuscript, great work.! Please note my comment about image acquisition.

---

## Round 0.3 · accepted · Accept

The response provided by the authors sufficiently addresses the concern raised by the reviewer regarding image acquisition. The manuscript adds significant value to global coral monitoring and warrants publication. Congratulations to the authors!